# High Added-Value by-Products from Biomass: A Case Study Unveiling Opportunities for Strengthening the Agroindustry Value Chain

**Filipe Kayodè Felisberto Dos Santos ***, **Ian Gardel Carvalho Barcellos-Silva, Odilon Leite-Barbosa, Rayssa Ribeiro, Yasmin Cunha-Silva *** **and Valdir Florencio Veiga-Junior ***

Chemical Engineering Section, Military Institute of Engineering, Praça General Tibúrcio, 80,
Rio de Janeiro 22290-270, Brazil; odilonleitebc@gmail.com (O.L.-B.); rayssaribeiro_92@hotmail.com (R.R.)
* Correspondence: filipe.kayode@gmail.com (F.K.F.D.S.); ycunhasilva@gmail.com (Y.C.-S.);
valdir.veiga@gmail.com (V.F.V.-J.)

**Abstract:** The current era witnesses a remarkable advancement in biomass utilization, guided by the principles of green chemistry and biorefinery and the comprehensive exploitation of plant-based raw materials. Predominantly, large-scale production methods have been pursued, akin to approaches in the oil industry, enabling the incorporation of novel products into energy and petrochemical markets. However, the viability of such systems on a small and medium scale is hindered by logistical challenges and the constraints of economies of scale. For small agricultural producers and food processing companies, the complete utilization of biomass transcends environmental responsibility, evolving into a strategy for survival through the diversification of by-products with enhanced value. The state of Rio de Janeiro in Brazil presents a range of population dynamics, geographical features, climate conditions, and agricultural production patterns that closely resemble those found in various tropical countries and agricultural regions worldwide. This region, sustaining a green belt supporting 17 million people, provides an apt case study for investigating chemical compounds with potential value among agro-industrial residues, which can motivate the creation of a lucrative biotechnological industry. Examples include naringenin and hesperidin from oranges and lemons, epi-gallo-catechin gallate from bananas, caffeic acids from coffee, and the bromelain enzyme from pineapples. This study addresses the challenges associated with developing biotechnological alternatives within the agroindustry, considering economic, technological, logistical, and market-related aspects. The insights from examining the Brazilian state of Rio de Janeiro will contribute to the broader discourse on sustainable biomass utilization and the creation of value-added by-products.

**Keywords:** high-value phytochemicals; biomass by-products; agroindustrial wastes

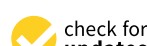



## 1. Introduction

The expansion of the food supply has been a longstanding global discourse, assuming heightened significance as the world's population exceeds 8 billion. Despite increased food production and consumption, widespread access to food remains hindered by issues, such as poverty, conflicts, natural resource mismanagement, and the emerging impact of climate change [1]. The year 2020 witnessed a remarkable milestone in global agricultural production, with the USA and China emerging as leading contributors, collectively reaching 4 billion tons. However, a disconcerting reality shadowed this achievement, as approximately 30% of this substantial food output, equivalent to 1.3 billion tons and incurring estimated costs exceeding 1 trillion dollars, succumbed to wastage. The United Nations Environment Programme attributes this loss primarily to transportation inefficiencies (40%) and mismanagement within restaurants and households (60%) [2]. Despite the surge in production, there remains a conspicuous lack of improvement in the living conditions of small- and medium-sized farmers, hindering the potential evolution of the

bioeconomy. The exploration of alternative applications for agricultural products and the strategic transformation of their waste into valuable by-products offer a promising avenue to revitalize production chains. This approach not only fortifies value chains but also champions sustainability in agriculture, ensuring the enduring presence of individuals in rural areas [3,4].

Efficiently harnessing agricultural waste has become a focal point in sustainable practices, notably within the framework of biorefineries. Inspired by the principles of petroleum refining, these models strive for the comprehensive utilization of biomass, separating valuable fractions for applications beyond mere energy production. Medium-scale productions concentrated in small regions benefit from logistical advantages but can only thrive when focusing on higher-value products. For energy production, scale gains are crucial for viability. The selective extraction of high-value bioactives, such as enzymes, antioxidants, and flavorings, from agroindustry waste presents a promising avenue for establishing new, intensive biotechnology industries. These endeavors could flourish around metropolitan agricultural belts and regions characterized by concentrated agriculture [5–9].

One noteworthy aspect is the potential of these residues within the chemical domain, showcasing the ability to extract compounds of significant relevance for the cosmetic, pharmaceutical, textile, cleaning product, and functional food markets. Residues stemming from the grape juice and wine industry, for example, encompass a diverse array of compounds, including vitamin E found in the seeds. Post-extraction, vitamin E levels in the oil can vary between 1 to 53 mg per 100 g [10]. Within the wine sediment, the existence of potassium bitartrate and calcium tartrate facilitates purification, yielding cream of tartar and Rochelle salt. This process generates four distinct by-products intended for application across the food, beverage, textile, and metallurgical industries [11]. Numerous studies have explored the valorization of commodity residues, exemplified in the production of concentrated orange juice. This process yields essential oils, flavonoids, pectin, semi-crystalline cellulose, and by-products with heightened added value. In addition, it also results in high-value hydrolytic and oxidative enzymes crucial for the degradation of lignocellulosic materials, including laccase, and xylanase [12].

The state of Rio de Janeiro, Brazil, presents one of the largest green belts in Latin America, producing 170 million tons in 2020. With its diverse topography, climates, and tropical biomes, the state accommodates a rich tapestry of crops, catering to the needs of a population of 17 million inhabitants. This agricultural production generates thousands of tons of waste and cooperates with several food processing industries that also have challenges in managing industrial wastes. Furthermore, the region hosts a multitude of chemistry and biotechnological research institutions that develop methodologies to use the regional biomass but, indeed, still are not well economically supported. This paper utilizes this state as a case study, given its striking topographic, populational, and climate resemblance to many countries across Latin America, the Caribbean, Africa, Asia, and Oceania, characterized by intensive tropical agricultural production and the generation of residues ripe for exploitation through existing biotechnologies. Our examination encompasses local productions and agricultural and industrial processing residues and identifies substances and technologies conducive to establishing a biotechnological hub around this agroindustry. Logistics, the market, and technology gaps will also be assessed, highlighting the main strategies to overcome the different array of challenges. Such a hub has the potential to support small- and medium-sized agriculture (mainly peasant farmers) bolstering the value chain and allowing for a sustainable local bioeconomy.

## 2. Rio de Janeiro State: A Case Study

### 2.1. Biodiversity and Climatic Patterns in Varied Vegetation Landscapes

The State of Rio de Janeiro is located in the Brazilian South-East region. It extends between latitudes 20.5 and 23.5° S (approximately 300 km north-south) and longitudes 41 and 45° W (about 400 km west-east), covering an area of 43,780.157 km$^2$ (Figure 1A). It is

characterized by a highly diversified climate due to its rugged topography, featuring hills, mountain ranges, valleys, diverse vegetation, lowland areas, bays, and proximity to the Atlantic Ocean. Its latitudinal position provides ample exposure to solar radiation. In terms of the spatial distribution of air temperature and precipitation, the prominent presence of the Serra do Mar, locally known as Serra dos Órgãos, is a sea mountain range that stands out, with altitudes ranging from 100 to 2275 m. These geographical features contribute to the construction of the biodiversity in Rio de Janeiro, giving rise to ecosystems, such as mangroves and restingas (coastal sandy plain vegetation), high-altitude grasslands, and an extensive array of tropical dense ombrophilous forest formations—with the Atlantic rainforest, known as Mata Atlântica, being the principal biome that encompasses the entire state [13].

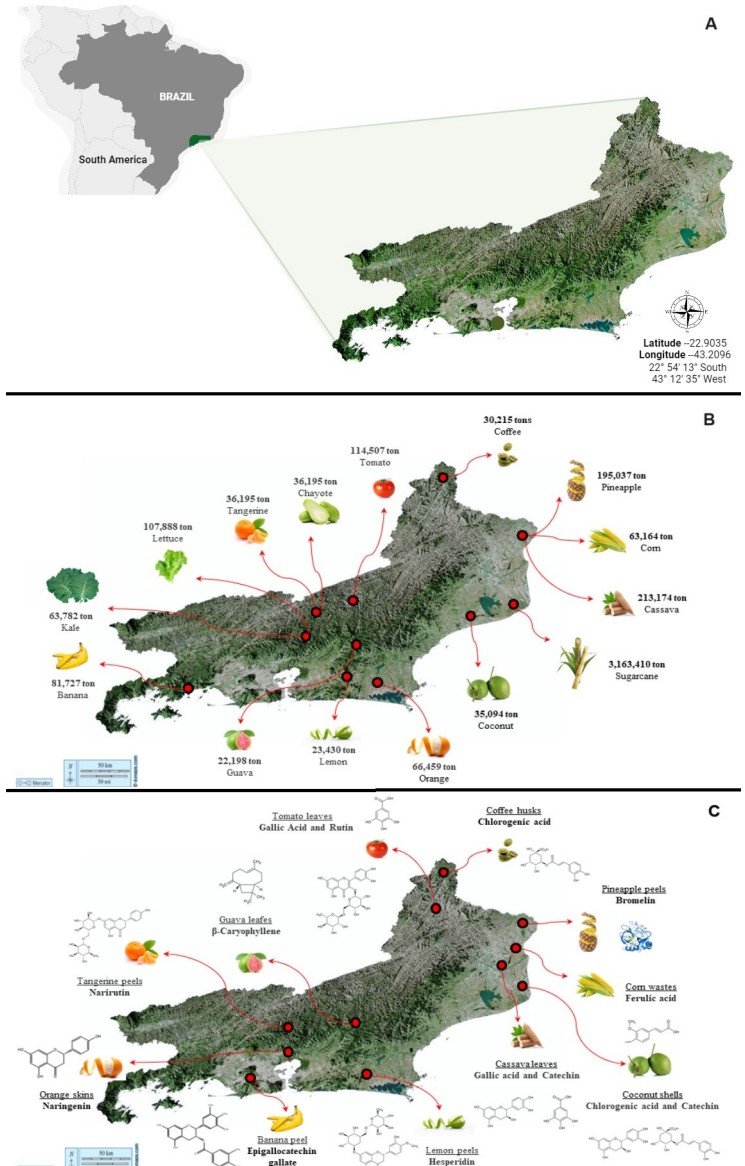

**Figure 1.** (**A**) The Rio de Janeiro State; (**B**) main agricultural production of the state [14]; (**C**) main high-value substances present at the agroindustrial wastes.

Such diverse tropical climates and vegetation allow for different crop production in each of these ecosystems, substantially contributing to varied environmental, economic, and social development. The agricultural production of the State comprises 66 different products, with a notable emphasis on sugarcane, cassava, pineapple, tomato, lettuce,

chayote, banana, orange, kale, and corn, collectively constituting approximately 90% of the total output. The regions with the highest agricultural production in the State of Rio de Janeiro are the northern (67.6%), central (14.7%), and mountainous regions (12.5%) (Figure 1B). The flat and hot northern region stands out for the huge production of sugarcane, intended for ethanol conversion. The central flat, hot, and rainy region, close to the mountains, is characterized by major productions of cassava and fruits. In the mountains, the green belt of the state has the most diversified production, from the altitude production of tomato and coffee, and also all horticultural production [14].

The mountainous region of the State of Rio de Janeiro is characterized by plateaus with relatively high altitudes, a factor influencing a mild climate with summer averages of 20 °C and winter averages of 15 °C. This region also exhibits areas with a high precipitation index (above 2000 mm/year) and several high-fertility valleys. These climatic conditions foster the cultivation of vegetables and fruits, including lettuce, chayote, coffee, tomato, kale, cabbage, broccoli, and cucumber. The fertility and geographical features of the elevation and climate bear resemblance to regions in Africa, such as the Bié Plateau located in central Angola and the Simien Mountains in southern Ethiopia; Latin America, such as Sierra Madre (México), Sierra Madre de Santa Marta (Colombia), and also Honduras, Guatemala, Ecuador and Peru; and also the Vietnã, Cambodia, and the Thailand region [15].

The northern region is characterized by the formation of a flat relief with lowlands, encompassing areas with altitude variations ranging from 50 m to 500 m. It includes stretches of semi-deciduous seasonal forests and areas with an extensive strip of sandy deposits parallel to the beach line, characterized by a hot and humid climate, with rainy seasons in the summer and dry periods in the winter [16]. These factors provide ideal conditions for the local production of sugarcane, pineapple, cassava, corn, and coconut, which constitute the primary production in the north. Pineapple, cassava, corn, and coconut production spans temperature ranges of 20 °C to 30 °C and an average annual rainfall of 1000–1500 mm, with higher temperatures and lower rainfall compared to the mountainous region. These aspects of relief, climate, and precipitation are akin to other regions, such as the Lockyer Valley, in Queensland, Australia, and Yucatán, México [17].

The central region of the State encompasses lake areas and the metropolitan zone of the main city, Rio de Janeiro. This geographical area displays low-amplitude relief features characterized by lowlands and flattened surfaces (50–120 m in elevation). The climatic aspects of this region are predominantly influenced by a warm and humid climate, marked by rainy seasons in the summer and dry periods in the winter. The average temperature along the coastal lowlands ranges from 22 °C to 23 °C annually, with winter lows reaching 18 °C and higher temperatures in the summer. Rainfall indices in the lowland regions can vary between 800 and 1300 mm in coastal areas [18]. The primary agricultural products include sugarcane, cassava, oranges, bananas, lemons, and guavas. Soil characteristics, such as a flat topography and minimal undulation, close to the mountain region, with several rivers crossing the region, make this region ideal for the cultivation of fruits, such as bananas and citrus, and also tubers, like cassava. Beyond the soil attributes, other factors, such as precipitation and temperature in the region, contribute to the feasibility of cultivating fruits. Citrus fruits thrive well in temperatures ranging from 21 °C to 32 °C with an average rainfall of 1200 mm. Banana crops are produced under optimal temperatures between 15 °C and 35 °C, with a rainfall range of 1200 to 2160 mm. Guava productions are well-developed within the temperature range of 25 °C to 28 °C and a precipitation range of 800 to 1000 mm. Similar climatic characteristics can be observed in Thailand, close to Bangkok [19].

## 2.2. Fruit and Vegetable Cultivation Belts

Fruits and vegetables cultivated in the State of Rio de Janeiro serve various purposes, including direct consumption, utilization in food industries, and exportation. Throughout the intricate processes involved, the generation of wastes is evident at distinct stages, such as harvesting, transportation, and processing. Notably, processing industries focused on

the production of pulps and juices play a pivotal role in contributing substantial amounts of residues. These residues encompass peels, seeds, stems, and fruits that are damaged or deemed unsuitable to consume but that can be used for the extraction of bioactives. Common fruits, like bananas, oranges, lemons, mango, and watermelon, collectively contribute to an annual waste output ranging from 25 to 57 million tons, with peels constituting the most voluminous discarded component, ranging from 15 to 60% [20,21].

The State of Rio de Janeiro is characterized by its predominant sugarcane production, covering 68% of the total output, equivalent to 3.16 million tons [14]. Various fruit crops are cultivated across the state's regions (Figure 1B), with notable productions, including pineapple (195 thousand tons (kt)), tomato (114 kt), chayote (103 kt), banana (82 kt), and orange (66 kt). Additionally, vegetables and tubers also contribute significantly to the agricultural output, with noteworthy productions of cassava (213 kt), lettuce (108 kt), and kale (64 kt). The production of crops, such as tomato, orange, banana, pineapple, corn, and cassava, finds employment in processing industries, with tomato used for extracts, orange for juice production, banana and pineapple for confectioneries, corn for canned goods, and cassava in tapioca starch.

### 2.3. Biomass Residues from Agroindustry and Plant Processing

Residues from agroindustrial processes can be generated at various stages of production, from cultivation and harvesting to industrial processing. These residues may consist of different parts of the plant and fruit, such as leaves, branches, peels, seeds, pulp, and residual water [22]. In the state of Rio de Janeiro, crops, like guava, orange, tangerine, and lemon, typify production, generating residues during cultivation, but also from harvesting and industrial processing (Figure 1C). The residues from cultivation originate from tree pruning aimed at managing growth and easing harvesting. Consequently, pruning yields tons of leaf and branch residues ripe for repurposing into valuable green products, such as high-quality essential oils. The market value of essential oils underscores their potential; for example, 10 mL of guava leaf essential oil reaches US$ 23.76. During the harvesting stage, certain crops, such as cassava, potatoes, carrots, and tomatoes, generate additional residues in the form of leaves. Leaves from cassava and tomato crops serve as sources of phenolics, such as rutin or kaempferol-3-O-rutinoside and rutin or gallic acid, respectively [23,24]. These phenolics found in leaf residues can be harnessed as antioxidants in the food industry. During the processing stage by industries to produce canned goods, juices, sweets, etc., residues, such as peels, seeds, stalks, and wastewater, are generated. The processing of citrus, pineapples, bananas, and coconuts is associated with the generation of high volumes of peels, which are rich in bioactives. Another byproduct of banana production associated with the generation of significant volumes of residues is the stalk, which comes from the banana bunch. These byproducts consisting of peels and stalks are sources of phenolic substances (hesperidin, narirutin, naringenin, epigallocatechin gallate, catechin, chlorogenic acid, and ferulic acid) (Figure 1C) that are of interest to cosmetic industries, dietary supplement, and food additive industries, etc. [25–30]. Additionally, there are industrial residues from fruit processing, such as tomato pulp from extract or juice production, which can be sources of lycopene, which can be used in cosmetics, natural food colorings, and dietary supplements [31,32]. The next part of this article will address four of the main crops produced in the State of Rio de Janeiro and their potential for generating high-value bioproducts with their respective waste that could strengthen production and value chains.

### 3. Waste Reuse

The utilization of renewable and environmentally sustainable agroindustrial residues as raw materials for the production of bio-based products has recently garnered significant interest. The high heterogeneity of plants and the chemical complexity inherent to biomass residues provide a rich source for diverse products, including food, chemicals, textiles, pharmaceuticals, cosmetics, and related applications. Among the vast biomass resources, noteworthy examples include oleaginous, saccharidic, starch-rich, and ligno-

cellulosic species. Oleaginous plants, such as soybean (*Glycine max*) and oil palm (*Elaeis guinensis*), represent a substantial volume of biomass. Saccharidic species, exemplified by sugarcane (*Saccharum* spp.) and sorghum (*Sorghum bicolor*), as well as starch-rich crops, like corn (*Zea mays*) and beets (*Beta* spp.), contribute significantly to the available biomass of agroindustrial residues, offering promising avenues for energy production. Additionally, various commercially utilized species, including timber species, serve as sources of ligno-cellulosic materials, such as oilseed cakes, bagasse, straw, and sawdust. These materials have been extensively studied using methodologies, like enzymatic degradation, pyrolysis, gasification, and fermentation, aimed at large-scale energy production and the generation of platform molecules for replacing petrochemicals. The investigation and application of these methods contribute to the sustainable utilization and the development of bio-based substitutes for petrochemicals in commodities [33,34].

The utilization of waste in biorefinery models presents numerous challenges. Small-to-medium-scale or fragmented production, as well as waste generated in remote locations from the recovery plant (typically within a 50 km radius), poses significant obstacles due to the costs associated with accumulation and local transportation logistics, including expenses related to trucks and personnel. The low density of waste exacerbates these challenges, resulting in large volumes with minimal weight. Additionally, the heterogeneous nature of raw materials presents further complexities. Few strategies can effectively utilize heterogeneous biomass, whether due to variations in the relative amounts of cellulose, hemicellulose, and lignin or catalytic difficulties. Consequently, when the final product yields low added value, such as in cases of energy utilization or for petrochemical applications, its viability on small-to-medium scales becomes infeasible, unable to compete with petrochemicals benefiting from established production and distribution chains. In practice, only a few raw materials prove to be viable for generating bioproducts in biorefinery models, unless they are commodities [35,36].

The comprehensive utilization of biomass residues, which often serve as commodities, offers a range of significant benefits for both the environment and the economy. This includes the provision of a new library of biomolecules known as platform molecules, crucial for advancements in biotechnology. In regions, like Rio de Janeiro, for instance, where sugarcane production is substantial (Figure 1B), particularly for fuel ethanol, biorefinery models are extensively deployed. These models integrate alternative energy production from second- and third-generation ethanol, as well as the utilization of lignins and other residues [37].

However, close to these large commodity producers, a considerable portion of the population relies on micro- and mini-agricultural producers, mainly peasant farmers. These smaller-scale producers face environmental challenges and lack technological solutions to utilize residual biomass effectively, thereby sustaining their presence in rural areas. For them, solutions that generate by-products with high added value are imperative. In such cases, where production diversity is broad but heterogeneous and operates on a smaller local or medium regional scale, exploring the specificities of each raw material to extract valuable substances becomes paramount. Among the macromolecule choices, proteins, enzymes, and pectin are highly sought-after by industries. Within the realm of specialized plant metabolites, previously known as secondary metabolites, there exists a range of alternatives, including highly antioxidant and anti-inflammatory phenolics, specific and bioactive terpenes, and potent alkaloids. In Europe, where rice, wine, and olive oil production span across various countries at the south of the continent, such as Portugal, Spain, France, Italy, and Greece, research has focused on alternatives, such as levulinic acid from rice straws, lactic acid from grapes, and xylose from olive seeds [38–40]. Tropical countries and regions, such as Rio de Janeiro State, with their diverse agricultural landscapes (Figure 1B), present additional perspectives (Figure 1C). The residues and possibilities of four of these fruits, really valuable raw materials with several known biotechnological processes already developed, will be explored below.

### 3.1. Pineapple (Ananas comosus)

Pineapple (*Ananas comosus*) stands as the most significant species within the bromeliad family (Bromeliaceae), with Costa Rica, the Philippines, Brazil, Thailand, and India being the leading producing countries. Residues generated from pineapple consumption, commercialization, harvesting, and processing practices can be categorized into two distinct types: agricultural residues (such as leaves, roots, and stems) and residues resulting from processing practices (including the crown, peel, and core). Currently, initiatives are underway to implement production and processing practices that enable the comprehensive utilization (via biorefinery) of pineapple, aiming to provide a wide range of value-added products derived from food waste processing for the food, cosmetic, and pharmaceutical industries [41].

The main residue from pineapple processing is its peel, accounting for about 37% of the total weight [41]. Despite its bulkiness, this peel holds immense potential for utilization since it is rich in an enzyme called bromelain, a protease with multiple industrial uses. With approximately 30% of the peel's mass, bromelain possesses high added value, with various commercial products commanding a market value higher than that of the fruit itself [42,43]. It is non-toxic and exhibits antibacterial, antiparasitic, and anti-inflammatory properties, offering benefits in alleviating symptoms of chronic degenerative conditions (Figure 2). Bromelain has also been shown to aid in the wound-healing process, oral health, and post-surgical recovery, including in cardiac diseases [44].

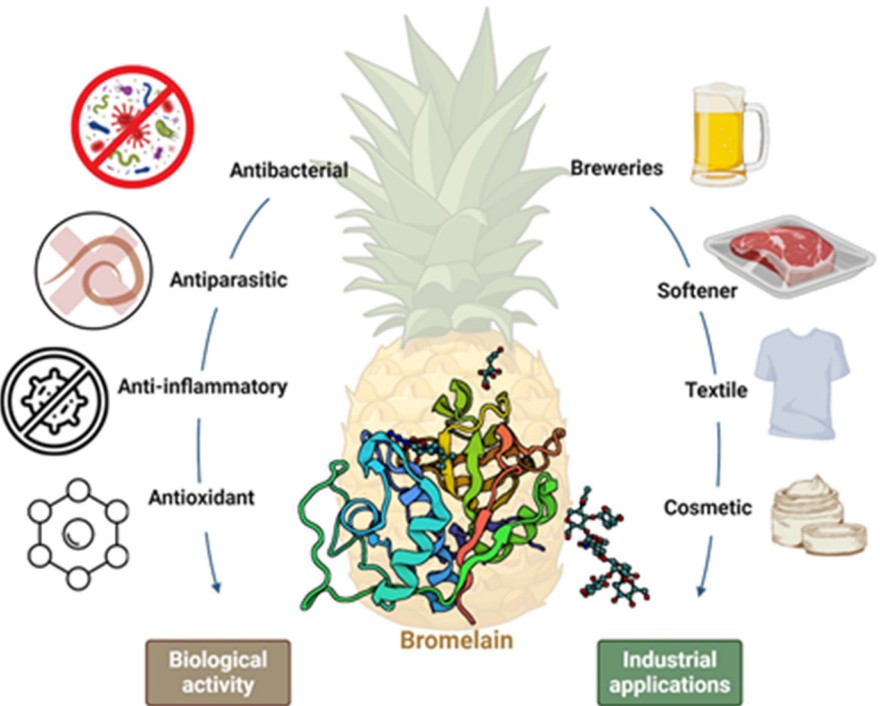

**Figure 2.** Bromelain uses and properties.

As a proteolytic enzyme, encompassing a variety of cysteine proteases, bromelain acts as a wide-spectrum protein digestant. For example, enhancing fibrinolytic activity (preventing blood clots) and reducing levels of bradykinin (a peptide causing inflammation) and fibrinogen (a glycoprotein complex involved in blood clot formation), thereby reducing excessive coagulation and regulating blood pressure and aiding in pain reduction and fluid retention (edema). It also acts to decrease the prostaglandin E2 (PGE2) and thromboxane A2 levels mediated by cyclooxygenase (COX), both associated with acute inflammation, and modulating specific cell surface adhesion molecules of immune cells [45].

In addition to its clinical applications, bromelain has also been utilized in various industries, such as food processing (in meat tenderization, hydrolyzing the meat myofibrils

of muscle protein), textiles (in leather processing and making silkworm cocoons softer, making the silk production process easier), cosmetics (wrinkle, acne, and dry skin treatment; tooth whitening), and in the formulation of detergents, to easily clean dirt containing proteins, such as meat broth [45].

With the advent of recombinant DNA technology, scientists and researchers worldwide have been exploring recombinant bromelain to achieve novel applications in the future. Efforts are underway to obtain highly purified bromelain in fewer steps and at a lower cost [46]. The actual extraction-of-bromelain process presents not only environmental appeal but also significant economic viability. The commercial product is obtained through a series of processes, including centrifugation, ultrafiltration, lyophilization, and high-performance liquid chromatography. Following extraction, the crude enzyme-containing mixture undergoes extensive purification to eliminate impurities, thus safeguarding its application efficacy and enzymatic activity. It is noteworthy that highly purified bromelain commands a market value of up to $2400 per kilogram [47]. Therefore, the extraction and purification strategies designed should be selective for high-yield purity, utilizing inexpensive inputs, with rapid and effective methodologies. Extraction typically occurs in buffered media at low temperatures where the extracted proteinases are purified using different methods, such as an aqueous two-phase system, chromatographic separation, reverse micellar system, and membrane filtration. These enzymes are highly important at the industrial level, with an annual sale of approximately $3 billion, owing to their wide application in food industries, pharmaceuticals, and detergent manufacturing [48].

Brazil ranks among the world's top pineapple producers, holding the fourth position, with 2.32 million tons grown across approximately 68.15 thousand hectares, contributing around $400 million to the national agricultural GDP. In 2022, the state of Rio de Janeiro yielded a total of 113,442 fruits across 4224 hectares, securing the fourth position nationally and generating approximately 62,960 tons of pineapple peel waste. Consequently, the amount of waste peels present is about 18,888 tons of bromelain, translating to an estimated value of $45.3 million [49].

The entire purification process is quite costly and time-consuming, involving relatively complex equipment commonly found in chemical and biotechnological research laboratories, such as centrifuges and chromatographs. However, in this model of bioactive production, purification processes can be segmented. The initial concentration stages can be carried out near the production site, while the final purification steps, requiring high technology, can be performed in nearby regions, as smaller volumes are more easily transported. The functionality of this process lies in the fact that climatic conditions allow nearby regions to also benefit from the biotechnological infrastructure for bioactive purification if they have similar crops, as usually happens. This is precisely the case with pineapple in northern Rio de Janeiro. In neighboring states, just over 100 km away, the production is up to 200 times greater, a billion-dollar possibility.

Another relevant characteristic is related to climate change. Climate variations strongly impact small- and medium-sized farmers, a phenomenon often referred to as climate vulnerability. Excess or lack of rainfall at critical times can often lead to crop failures, resulting in pulps becoming either too dry or excessively ripe, rendering them undesirable for consumption. Indeed, fluctuations in the water and sugar content of the pulp have relatively little effect on the constituents of the peels, making the utilization of their by-products viable for the entire fruit chain, as diversification stabilizes the production chain by strengthening the value chain. Thus, even with a pineapple pulp crop failure, the use of peels for the extraction of valuable bioactives, such as bromelain, maintains the viability of production.

*3.2. Banana (Musa spp.)*

The banana plant, a perennial fruit-bearing species, finds widespread cultivation across tropical and subtropical regions globally, ranking as the world's fourth most significant food crop, following rice, wheat, and maize. Predominantly grown in Asia, Latin America, and

Africa, India emerged as the largest producer in 2019, trailing China, Indonesia, Brazil, and Ecuador. Bananas reign as the most popular fruit, contributing to 16.8% of the total global fruit production. Over the past two decades, global banana production has witnessed a steady rise, surging from approximately 46 million tons in 1993 to around 119 million tons in 2019 [50,51]. With over 1000 varieties, bananas collectively amount to approximately 88 million tons imported annually for human consumption [52,53]. The fruit of the banana plant (*Musa* spp.) boasts a high nutritional content and is consumed fresh or processed into various products on small industrial scales, such as dried fruits, snacks, ice cream, bread, flour, spirits, and ingredients for functional foods. Bananas mainly stem from two species, *Musa acuminata* and *M. balbisiana*. *M. balbisiana* yields starchy bananas, commonly known as plantains or cooking bananas, typically consumed boiled and/or fried. On the other hand, the species *M. acuminata* produces the bananas commonly used in desserts and sweet dishes [54,55].

Despite the banana plant's fruit being a rich source of nutritional compounds, its cultivation yields significant amounts of waste. Banana leaves, leaf stems, peels, and pseudostems are among the primary residual biomass generated from banana cultivation [56]. With global banana exports valued at over $120 billion, exploring the by-products through the lens of green chemistry and the circular economy concept offers an opportunity to repurpose these wastes as valuable resources for the food, cosmetic, and pharmaceutical industries and for the development of new materials [57]. Banana peels constitute a significant portion of domestic and industrial food waste, accounting for approximately 35% of the total fresh mass of ripe fruits. Over the past two decades, Brazil has been the world's second-largest banana producer, contributing to a global production of approximately 6.6 million tons cultivated across 455 thousand hectares, generating revenues of around $2.7 billion annually. Roughly 53% of Brazil's banana production undergoes industrial processing, indicating the potential generation of 1.2 million tons of banana peel waste. In 2022, the State of Rio de Janeiro contributed 64,088 tons, resulting in an estimated waste generation of approximately 21,789 tons solely from the state [58]. With just 1% of the country's total banana production, the alternatives developed in Rio de Janeiro have the potential for significant expansion and a huge economic impact.

While various studies propose different strategies for utilizing banana biomass waste, there is still progress needed to achieve more efficient and truly sustainable processes. Banana peel waste, for instance, is rich in phenolic compounds, with a total phenolic content ranging from 4.95 to 47 mg/g dry matter. Compared to the peels of other fruits, banana ranks second in terms of the phenolic content. More than 40 phenolic compounds have been identified in banana peels, with flavanols being the predominant group among these compounds [58–61]. One of the most substantial portions of banana waste is the stem (peduncle), the central part of the bunch (inflorescence) where the amount of palms (cluster) is attached. The stem is rich in a flavonoids belonging to the catechin class, such as epigallocatechin gallate (EGCG) (Figure 3), an antioxidant, anti-inflammatory, and anticancer substance, used for skin lesions and for reducing Alzheimer's disease pathology [62–64].

EGCG was also already described to aid in the treatment of metabolic syndrome, obesity, diabetes, and hypertension [65]. Its commercial value is remarkable, with a market price of approximately US$ 200 for each 50 mg.

In Rio de Janeiro's central region, close to the mountain regions, several food processing industries produce banana sweets for the internal market and export to Canada, CEE, and Japan. The scale of them is small. Considering only one of these food industries, it generates around 2 tons of peel/stem waste daily, totaling approximately 40 tons per month. Extracting about 0.01% of EGCG from these residues yields about 4 kg of EGCG, generating an income of over US$ 16 million per month. However, to effectively harness these benefits, further research is needed to fully understand the chemistry and biological activity of these compounds. Additionally, it is crucial to develop efficient and sustainable methods for recovering these secondary metabolites from banana waste [66].

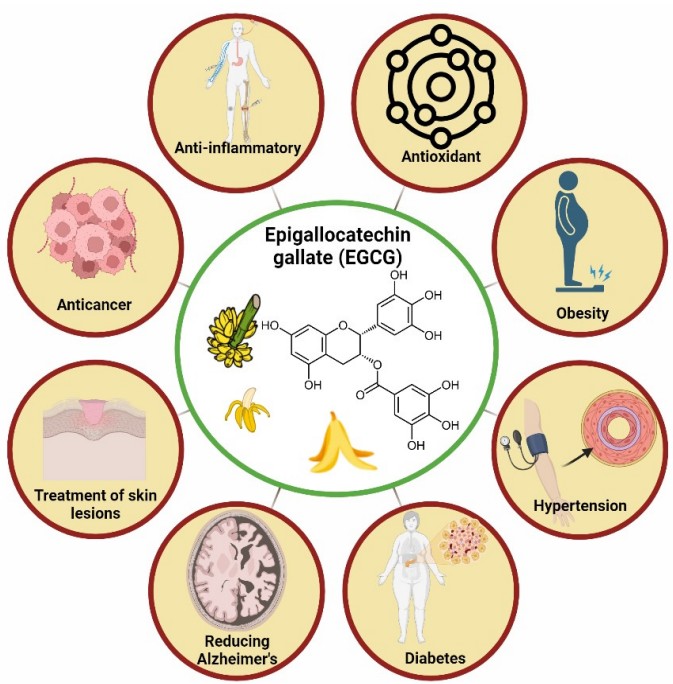

**Figure 3.** Flavonoids and epigallocatechin gallate use and properties.

Among the phenolics present in banana wastes, the flavonoid rutin was also identified in a high concentration, of 973.08 mg/100 g in the ethanolic extract of banana peel [67], as observed in Table 1.

**Table 1.** Identified compounds in banana peel.

| Phenolic Compounds | Banana Peel | Quantity | References |
|---|---|---|---|
| Kaempferol | Red banana | 28.80 µg/mL | [68] |
| | Yellow banana | 9.30 µg/mL | [68] |
| Isoquercitrin | Red banana | 14.54 µg/mL | [68] |
| | Yellow banana | 10.47 µg/mL | [68] |
| Rutin | | 973.08 mg/100 g | |
| Myricetin | *M. paradisiaca* | 11.52 mg/100 g | [69] |
| Naringenin | | 8.47 mg/100 g | |
| Ferulic acid | *Musa* spp. | 1.63 mg/100 g | [70] |
| | | 60 mg/100 g | |
| Cinnamic acid | | 1.93 ng/g | |
| Alpha-hydroxycinnamic acid | Karpooravalli | 40.66 ng/g | [71] |
| Sinapic acid | | 10.29 ng/g | |
| p-Coumaric acid | | 8.05 ng/g | |
| Dopamine | Grande Naine | 1.72 mg/g | |
| | Gruesa | 1.17 mg/g | [72] |
| L-dopa | Grande Naine | 0.31 mg/g | |
| | Gruesa | 0.56 mg/g | |

Rutin discarded in banana peels can be found on specialized websites for the sale of reagents and chemicals, with prices reaching up to $500 for 50 mg (for example, www.sigmaaldrich.com (accessed 20 January 2024) [73], www.adooq.com (accessed 22 January 2024) [74], and www.biosynth.com (accessed 21 January 2024) [75]). Additionally, it exhibits various biological activities, such as strengthening capillary vessels, reducing symptoms of hemophilia, and preventing leg edema. Its deficiency can lead to the appearance of microvaricose veins and vascular problems. Thus, it has the potential to be used in pharmaceutical formulations and nutritional supplementation [76].

Another flavonoid of interest found in banana peels, which usually ends up as waste, is quercetin. This compound can have concentrations ranging from 3.2 mg/g to 12.5 mg/g in banana peels, is very ubiquitous, and presents similar concentrations in all banana

cultivars. Like rutin, its properties involve improvements in vascular function (significant reduction in blood pressure, decrease in triglyceride levels, and antioxidant activity). These compounds have high commercial values, ranging from $100 to $514 per 10 mg of sample [67]. Based on the same calculation applied to the generation of banana waste in the State of Rio de Janeiro, it would be possible to obtain around $4 million in rutin and $20 million per month.

### 3.3. Coffee (Coffea spp.)

Coffee is one of the most well-known and consumed products globally. According to the International Coffee Organization, Brazil is the largest producer and exporter and the second largest consumer of coffee [77]. The primary form of consumption is through the beverage produced from the roasted beans of the coffee plant, a shrub of the Rubiaceae family and *Coffea* L. genus. Nowadays, coffee is primarily consumed by infusing the beans of the *C. arabica* and *C. canephora* species [78]. The global coffee production (2023–2024 harvest) is estimated at 174.3 million 60 kg bags. Approximately 96.3 million bags are of the *C. arabica* species (55.2% of the world volume), and 78 million bags are of the *C. canephora* species (Robusta + Conilon types), which accounts for 44.8% of the world harvest {77}. Coffee consumption is high worldwide due to the presence of caffeine, a chemical compound found in larger quantities in coffee, which is isolated for use in energy drinks, pharmaceuticals, and cosmetics [79].

One of the primary biomasses generated in the preliminary processes of post-harvest coffee bean treatment, i.e., before they undergo the roasting process, is the silver skin. It is a thin outer layer covering the green coffee beans, which, after a washing process that prepares the beans for roasting, is separated in industrial processes via an airstream. Compared to other coffee by-products, the silver skin is relatively stable due to its low moisture content (5–10%), making it an intriguing product for various applications due to its longer shelf life [80,81]. The silver skin accounts for approximately 4.2% ($w/w$) of coffee beans [82]. Around 60 kg of silver skin is produced for every eight tons of roasted coffee [83]. In 2022, Brazil's coffee production reached 3,172,562 tons, with the state of Rio de Janeiro contributing 19,322 tons. This resulted in an estimated 144 tons of silver skin residue [84]. The vast array of bioactive substances present in the silver skin is reported to potentially aid in the treatment of diseases associated with the imbalance of reactive oxygen species (ROS). Besides possessing antioxidant activity, they are noted for their glycoregulatory properties, stimulant action, and efficacy as prebiotics [85,86]. The main constituents include chlorogenic acids (1–6%), caffeine (0.8–1.25%), and melanoidins (17–23%), which may vary due to geographical factors. Chlorogenic acids comprise two groups: those derived from hydroxybenzoic acid and those derived from hydroxycinnamic acid. Hydroxybenzoic acids include gallic, p-hydroxybenzoic, protocatechuic, vanillic, and syringic acids. Hydroxycinnamic acids, on the other hand, are aromatic compounds with three carbon atoms forming a side chain, such as caffeic, ferulic, and p-coumaric acids [87].

Among chlorogenic acids, the most prevalent bioactive compound in silver skin is 5-O-caffeoylquinic acid (5-CQA), extensively studied for its biological effects. It acts as a potent antioxidant and anti-inflammatory agent, with potential benefits in reducing Alzheimer's, Parkinson's, cardiovascular, cancerous, and metabolic syndrome diseases [88,89]. Commercially known as neochlorogenic acid, 5-CQA is priced at $915.00 for 50 mg and is typically present at a concentration of 52.53 mg/100 g in silver skin [90,91]. Considering the annual residual quantity of silver skin in the State of Rio de Janeiro (144 tons), repurposing this waste could yield approximately 75 tons of 5-CQA, translating into a potential income of around US$1.3 billion.

The technological processes required to isolate (or simply concentrate) chlorogenic acids typically involve chromatographic methods, particularly high-performance liquid chromatography (HPLC) (Figure 4). Together with detection systems that operate with mass spectrometry (MS), this technique greatly evolved in the last decades.

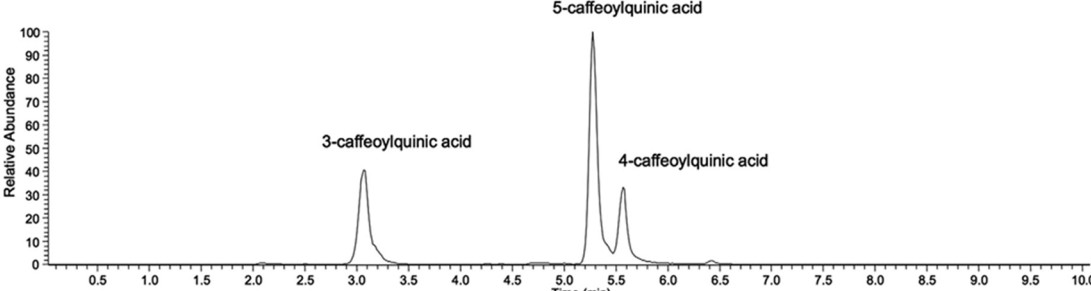

**Figure 4.** Identification of CQAs present in coffee silver skin via LC-MS: 3-caffeoylquinic acid, 4-caffeoylquinic acid, and 5-caffeoylquinic acid [92].

HPLC systems that operate in preparative mode, that is, with the capacity to separate large quantities of material, previously prohibitively expensive, have significantly improved their cost–benefit ratio, especially with the advent of Asian suppliers. Dedicated HPLC equipment is currently found by the dozens in every pharmaceutical and food factory. Furthermore, these systems allow for online use to purify a substance and also concentrate different fractions of other similar or different substances, greatly improving the purification or concentration processes, especially when it comes to matrices containing different bioactives of interest, such as the phenolic from banana wastes. In addition to CQAs, furanoid diterpenic acids used as a quality control pattern of coffee, among several other substances, are present in coffee wastes and could be used as valuable by-products.

Caffeic acid, a metabolite derived from chlorogenic acid, is also described in the literature as a phenolic compound found in coffee's silver skin (Figure 5). Despite possessing a lower commercial value compared to CQAs (1 g of caffeic acid equivalent to $50), it boasts a wide range of biological activities, including antioxidant, antiviral, anticancer, antiplasmodial, and antidiabetic properties [93–95].

With a high volume (144 tons) of silver coffee skin obtained from waste produced in the State of Rio de Janeiro, it is possible to estimate 14.4 million dollars per year in earnings from caffeic acid alone, considering that this compound represents 0.2% of dry material [96].

The significance of caffeic acid and its derivatives extends to the cosmetic industry, where it is valued for its antioxidant properties and its incorporation into formulations for skin and hair care. The reuse of compounds sourced from wastes aligns with the contemporary upcycling trend, which emphasizes the reuse of by-products and waste reduction. Consumers are more aware of the origin of cosmetic products and their environmental impact, promoting the phenomenon of upcycling [97].

### 3.4. Citrus (Citrus spp.)

Citrus fruits, belonging to the *Citrus* genus of the Rutaceae family, constitute a vital and diverse component of the plant kingdom. This taxonomic group encompasses trees, shrubs, and herbs distributed globally, with a significant presence in tropical, subtropical, and Mediterranean climate regions, such as Australia, California, Florida, South Africa, Italy, and Spain. Among the most renowned and commercially preferred citrus species are orange (*Citrus sinensis*), grapefruit (*Citrus maxima*), lemon (*Citrus limon*), lime (*Citrus aurantiifolia*), citron (*Citrus medica*), tangerine (*Citrus reticulata*), and pomelo (*Citrus paradisi*). These fruits, besides being the most cultivated and traded worldwide, are esteemed for their extensive array of applications, ranging from culinary uses, owing to their distinct flavor profiles, to pharmaceutical applications, attributable to their nutraceutical compounds, such as vitamins, phenolic compounds, and flavonoids [98].

**Figure 5.** The main value-added compounds present in coffee silver skin.

The global production of citrus fruits has experienced steady growth over the years, reaching a pinnacle of around 144 million tons in 2019. This surge reflects the escalating demand for citrus fruits, renowned for their health-enhancing attributes, as abundant sources of vitamin C and antioxidants. Among the largest producers, China (37.7 million tonnes (Mt)), Brazil (19.7 Mt), India (13.3 Mt), Mexico (8.4 Mt), and Spain (6.0 Mt) stand out, with significant productions contributing to the global supply. Brazil, notably, holds a preeminent position in orange cultivation, a cornerstone of the concentrated juice industry. The escalating production and consumption of citrus fruits worldwide not only underscore dietary preferences but also underscore the pivotal role these fruits play in the global agricultural economy [99]. On the other hand, it also resulted in huge amounts of solid waste.

Regarding the global processing of citrus fruits, Brazil (10 Mt) and the United States of America (3 Mt) stand out as having the largest volumes of processed fruits in 2019 [99]. During processing, the citrus industry generates two distinct types of by-products: a solid/semi-solid fraction consisting of fruit peels and residues and citrus wastewater. The magnitude of wastewater produced in this industry is considerable, estimated to range between 1.5 and 17 cubic meters per ton of fruit processed [100]. Further data indicate that industrial processes yield approximately 120 Mt of waste annually. Among these, only about 45% of the total fruit weight is utilized, leaving 55% as waste, comprising the peel, pulp, and seeds. This waste composition comprises 27% peel, 26% pulp, and

2% seeds. Often inadequately disposed of, these wastes contribute to environmental pollution and pose challenges for waste management and environmental sustainability efforts. The substantial variation in the liquid waste volume is directly linked to diverse processing methods employed in citrus industries, including varying technologies for juice and essential oil extraction, as well as the type of fruit processed [100,101].

The main chemical substances that can be extracted from citrus waste to become high-value-added bioproducts are essential oils and phenolics. Among the phenolics, rutin, naringenin, narirutin, and hesperidin stand out. Essential oils are primarily extracted from citrus peels, commanding significant market value that varies among different citrus varieties. Each type of orange, lemon, lime, or tangerine yields its distinctive essential oil, characterized by unique compositions and yields subject to fluctuations influenced by both biotic and abiotic factors. Citrus essential oils serve diverse industrial purposes in food, beverages, and hygiene products due to their antimicrobial and aromatic properties. Notable examples include limonene, a widely-employed monoterpenoid acting as a green solvent in the industry, alongside carvone and carveol. These substances derived from citrus waste present a sustainable and economically advantageous opportunity for the industry, facilitating the utilization of by-products that would otherwise be discarded [102].

In Rio de Janeiro, the primary citrus crops include lemon, orange, and tangerine. According to IBGE, in 2022, the fruit production comprised 63,683 tons of oranges, 33,799 tons of tangerines, and 21,575 tons of lemons. This totals 119,057 tons of citrus fruits produced, resulting in an estimated 32,145 tons of peel waste, including 17,194 tons of orange peels, 9126 tons of tangerine peels, and 5825 tons of lemon peels. From the peel waste, valuable essential oils can be extracted, fetching various market prices. For instance, 10 mL of orange essential oil (*C. sinensis*) sells for US$ 7.61, 10 mL of tangerine essential oil (*C. reticulata*) for US$ 7.65, 10 mL of Sicilian Lemon essential oil (*C. limon*) for US$ 9.47, and 10 mL of Tahiti Lemon essential oil (*C. aurantifolia*) for US$ 8.58. Therefore, taking into account the generated waste values, the mass-to-volume ratios of oil production, and the market value of essential oils, the reuse of citrus fruit peel biomass from the State of Rio de Janeiro could potentially yield US$ 130.8 million from orange essential oil production, US$ 33.3 million from Tahiti Lemon essential oils, and US$ 27.9 million from tangerine essential oils [103].

Pectin is another product that can be obtained from the peels of citrus fruits. The peels of citrus fruits are divided into two parts, the outer peel (flavedo or epicarp) and the inner peel (albedo or mesocarp), with the essential oil concentrated in the flavedo and the pectin in the albedo [104]. According to data related to oranges, the flavedo represents approximately 8–10% and the albedo 15–25% of the fruit [105]. Regarding the albedo of citrus fruits as waste, there is the possibility of applying these residues as sources of pectin for the food, cosmetic, and pharmaceutical industries. In the food industry, pectin is used as a gelling agent, texturizer, and emulsifier and for food encapsulation and coating. On the other hand, in the pharmaceutical sector, pectin is applied in drug delivery, probiotic functions, wound healing products, and tissue engineering [106–108].

Pectin is a heteropolysaccharide found in plants and fruits, being one of the main components of the cell wall. Currently, commercial pectin production is mainly obtained from beet, citrus fruit, and apple residues [108]. In this context, it is estimated that approximately 85% of the world's commercial pectin is produced from citrus fruit peel residues. From these residues, different levels of pectin can be obtained from citrus fruits, with peels of fresh fruits generally containing between 1.5% and 3% pectin, while dried peels can have higher concentrations, reaching about 9% to 18% pectin. Additionally, studies aiming to concentrate this substance have shown promising results, achieving yields of up to 28% pectin through the use of microwave-assisted extraction [109].

Considering that the production in Rio de Janeiro in 2022 was 63,683 tons of oranges, 33,799 tons of tangerines, and 21,575 tons of lemons, this totals 119,057 tons of citrus fruits produced. From this total, approximately 17,858–29,764 tons of citrus fruit albedo can be generated (9552–15,920 tons of orange residues, 5069–8499 tons of tangerine residues, and 3236–5393 tons of lemon residues). Starting from 17,858–29,764 tons of albedo, ap-

proximately 1607–2678 tons of pectin can be obtained from an extract with a 9% pectin content and 3214–5357 tons of pectin from an extract with an 18% pectin content [110]. Thus, considering the amount of pectin that can be generated from the residues of the State of Rio de Janeiro and the market value of $682.64 (per 1 kg), the reuse of biomass from citrus fruit peel albedo in the state could potentially yield $1.0–3.6 billion dollars [111].

In addition to essential oils and pectin, peels can be a source of flavonoids. This is the case for hesperidin, highly sought-after as a key ingredient in dietary supplements, cosmetics, natural food additives, and various pharmaceutical and nutraceutical applications [112]. Its diverse array of beneficial properties includes antioxidant, anti-inflammatory, antiproliferative, antiallergic, antiviral, anticancer, neuroprotective, and antimicrobial effects (Figure 6). Numerous studies indicate that hesperidin is the most prevalent flavonoid in citrus waste, boasting significant potential for conversion into high-value bioproducts. Recently published studies have documented hesperidin concentrations of $2298.99 \pm 190.70$ mg/100 g DW in orange peel, $1896.73 \pm 78.46$ mg/100 g DW in tangerine peel, and $24.51 \pm 0.18$ mg/g DW in lemon peel [113–115].

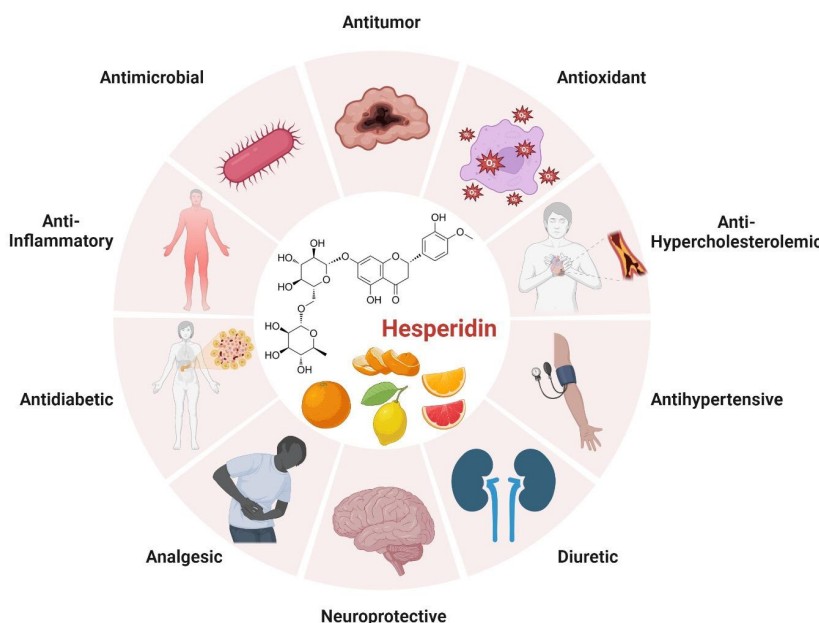

**Figure 6.** Hesperidin in citrus: uses and properties.

Alongside hesperidin, the heterocyclic bioflavonoid naringenin stands out as another valuable member of the polyphenol class. It has garnered significant attention due to its remarkable potential applications and wide range of biological activities. The efficacy in treating neurodegenerative diseases, obesity, arteriosclerosis, and various other chronic conditions has already been demonstrated scientifically. Extracting this bioproduct from natural sources, particularly agroindustrial by-products, offers a sustainable approach to valorizing waste, thereby contributing to circular economy practices within the agroindustry. The ongoing refinement of extraction and purification methods for naringenin, along with the exploration of innovative applications across fields, like pharmacology and biomedicine, underscores the burgeoning interest in high-value bioproducts [116,117].

From the State of Rio de Janeiro's production, the peel waste generated in 2022 (flavedo + albedo) amounts to approximately 14,647–22,289 tons for orange, 7773–11,829 tons for tangerine, and 4962–7551 tons for lemon. Of this total, about 605.8–921.8 tons of hesperidin from citrus fruits can be produced (336.7–512.4 tons resulting from orange waste, 147.4–224.3 tons resulting from tangerine waste, and 121.6–185.0 tons resulting from lemon waste). Thus, considering the total hesperidin that can be produced from the peels of the Rio de Janeiro State production and the market value of this product of US$ 456.02 (20 mg),

the reuse of citrus fruit peel biomass in the state could potentially yield 13.6–20.9 trillion dollars for the local economy [118].

Indeed, achieving the complete extraction of all flavonoids is quite challenging. Additionally, not all the peels would be accessible, making the logistics for extraction feasible. Ultimately, if a substantial quantity of these substances was available, it could lower the market price globally. However, even if the revenue was only 0.01% of the potential value, solely for the State of Rio de Janeiro, with its relatively small production compared to the entire country, the financial return would still be highly appealing.

The transformation of citrus waste into several high-value bioproducts presents a compelling economic opportunity within the agroindustry. By extrapolating these concentrations to the total waste volume in the State of Rio de Janeiro and across the country, which stands as the world's largest citrus producer, we derive a substantial monetary value estimate when multiplied by their respective market values. However, it is crucial to acknowledge that the actual quantity of extracted substances may fluctuate based on variables, like the citrus species, cultivation conditions, and the specific extraction methods utilized.

In the citrus industry, beyond concentrated juices, there is a plethora of products with significant market value, ranging from water to essential oils, flavonoids, and pectin, all of which can be extracted sequentially. Even after these extractions, there are still residues that can be utilized in biorefinery processes for energy production, further enhancing the production chain in alignment with circular economy principles and sustainability practices. This approach is not just about valorizing previously discarded waste; it also makes substantial contributions to innovation and sustainable development in agriculture. If it proves to be successful in Rio de Janeiro, it undoubtedly holds promise for any citrus-producing region worldwide.

## 4. Challenges in the Production Chain

Waste management in the production chain faces considerable challenges, especially concerning the collection, treatment, and valorization of these materials (Figure 7). The effective valorization of food and agricultural waste is crucial for reducing the environmental impact and improving sustainability. However, this requires robust infrastructure and optimized processes to ensure that waste is collected, treated, and transformed efficiently, minimizing degradation and maximizing its utilization potential [119]. Transforming waste into value-added products presents significant challenges, extending beyond technical issues. Some studies highlight the opportunity to convert agroindustrial waste into useful products for the industry, such as the sustainable management of coffee by-products, the potential use of pineapple processing waste, and the utilization of bioactive compounds present in these types of waste, such as apple, banana, mango, potato, carrot, beetroot, and many others. However, achieving this transformation involves not only developing appropriate technologies but also addressing regulatory, economic, and market acceptance barriers. These challenges require innovative solutions to ensure that waste is utilized sustainably and profitably [120,121].

Essentially, there are several critical areas in the valorization of urban solid waste, including food waste quantification, technological aspects, logistics and supply chain management, market demand analysis, social impact assessment, and policy and legislative considerations. Concerning logistics, market dynamics, and technological advancements, particular attention is given to the complexities and challenges associated with efficiently managing the supply chain, fostering sustainable markets for products derived from solid waste, and advancing technology to optimize the valorization process. These components play a pivotal role in the successful implementation of circular economy strategies involving urban solid waste [122].

## 4.1. Logistics

The logistics required for valorizing agroindustrial waste pose a multifaceted challenge, encompassing tasks from collection and transportation to processing. Although some of these wastes contain valuable components, the costs associated with their collection, transportation and processing are high. Consequently, many of these by-products are discarded, adding significantly to global waste production. The sustainable management of these residues is crucial, necessitating a shift from a linear "produce-use-dispose" approach to circular models of waste valorization, where residues are transformed into valuable commodities [122].

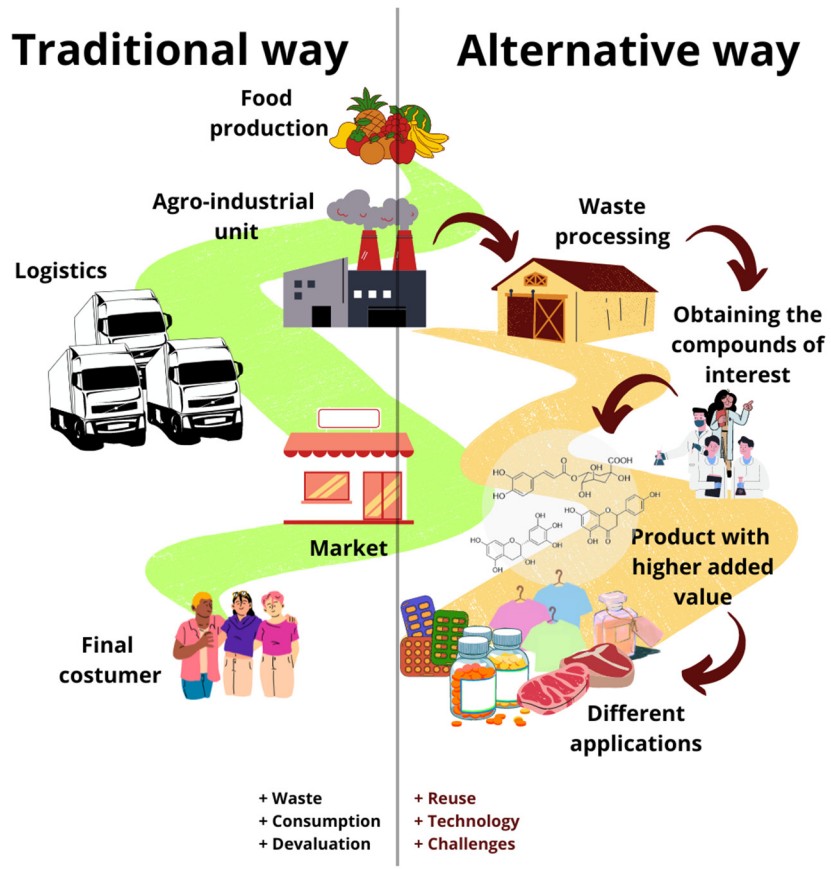

**Figure 7.** Obtaining value-added products from waste: an alternative in the agroindustry.

In Indonesia, the logistical management of food waste encounters significant challenges. Currently, the collection and processing of these residues are suboptimal, with the majority being directly disposed of in landfills and dumpsites. Only a small proportion undergoes collection and is repurposed for animal feed and composting. Issues stemming from improper disposal include pest infestations, foul odors, and harmful emissions, like methane and carbon dioxide, which adversely affect soil, air, and water quality. Moreover, landfills are economically infeasible due to the high costs associated with land and equipment, coupled with their requirement for extensive land areas, resulting in adverse environmental impacts. Despite governmental initiatives aimed at managing domestic solid waste, including food waste, through reduction, reuse, and recycling, challenges persist, such as insufficient financial support, poorly managed or inadequate facilities, and a lack of municipal backing [123].

The supply chain involved in biomass conversion to high-value products is intricate, covering biomass production, logistics, product manufacturing, distribution, and end-user consumption. Successfully managing this chain demands particular attention to operational logistics and sustainability considerations, including challenges, like harvesting, collec-

tion, storage, transportation, pre-treatment methods, and overall supply system design. Logistic efficiency plays a pivotal role in the biomass supply network, as delays can lead to biomass degradation and environmental emissions. Hence, logistics firms must prioritize the time-sensitive collection, storage, and delivery of inputs. In the context of food waste repurposing, logistical aspects, such as the waste density, selling price, transportation expenses, seasonality, high moisture content, geographic dispersion, and low calorific values, need careful consideration. The conversion of food waste into bioenergy faces significant challenges in terms of the infrastructure and transportation. Consequently, a deep understanding of these pivotal challenges and thoughtful deliberation on supply logistics, design, and operation are imperative for successful implementation. Engaging stakeholders and producers in planning, designing, executing, and decision-making processes across all levels remains critical for project success [124].

One critical logistic factor in turning waste into profitable by-products is fostering collaboration among small-scale producers, enabling them to handle various stages of the logistics chain. This collaboration may involve tasks, like proper storage, pre-processing, and the consolidated transportation of materials that are of lower volume but higher value. In agricultural regions, such as those in Rio de Janeiro, characterized by well-developed road networks for the transportation of fruits and vegetables, organizing waste concentration is not only feasible but also beneficial. It is sought after by small-scale producers, small enterprises, and local municipalities, as they possess the manpower and infrastructure necessary for the development of new products.

Nevertheless, the leadership exhibited by certain production hubs, notably in banana and pineapple cultivation, fosters increased collaboration among stakeholders and greater interest in participation in Rio de Janeiro. These examples, representing materials with low density and susceptibility to rapid degradation, are just a glimpse of the broader spectrum. For instance, the coconut water bottling and tomato processing industries face similar challenges. Encouraging the aggregation of medium-sized enterprises, thus facilitating the utilization of waste from various producer associations, stands out as one viable solution. Another approach involves the diversification of small-scale enterprises, as observed in the case of banana candy producers who now export a range of banana products including guava, pineapple, and mango derivatives. Given the volume of waste generated, establishing small-scale, multi-purpose extraction plants in proximate locations emerges as a feasible option.

### 4.2. Market

The introduction of any new product to the market encounters numerous challenges, spanning from accessing distribution channels to gaining consumer acceptance. Early in the development process, significant economic barriers arise, including the implementation costs of infrastructure, production expenses, and the need for costly purification procedures. While sourcing raw materials from waste presents its own set of challenges, ensuring that they meet market demands in terms of quantity and consistent quality, with regards to chemical, physicochemical, and biochemical attributes, such as oxidation levels, moisture content, and sanitation, can prove to be even more demanding.

The transformation of agroindustrial waste into high-value-added products demands the implementation of appropriate business models to ensure a mutually advantageous scenario for all stakeholders involved. Addressing the equitable distribution of costs and benefits among various actors, as well as effectively managing material resources and knowledge dissemination, stands as critical challenges in this endeavor. Additionally, the organic farming sector, along with the processing and distribution of organic food, is showing an increasing preference for sustainable inputs, such as bioplastics and biofuels derived from organic agricultural waste. This trend reflects a growing consumer acceptance and willingness to invest in products derived from organic agricultural residues [125–127].

In terms of market acceptance, products derived from waste must demonstrate competitiveness in both quality and price to earn the trust and preference of consumers. These

challenges are further compounded by regulatory barriers that can impede access to potential markets for such innovative products. To effectively promote the valorization of agroindustrial waste, it is crucial not only to advance processing technologies but also to devise market strategies that underscore the environmental and nutritional benefits of these products. This approach will encourage their adoption by consumers and industries alike [126,127].

Addressing the challenges of meeting specific quality, sanitary, and market standards while navigating regulatory hurdles, the uncertainty regarding the pricing and acceptance of these products is notable. To effectively tackle these obstacles, it is advisable to collaborate with companies that handle much of the labor-intensive tasks, including packaging, marketing, sales, exportation, and legal matters. Consequently, the biotechnological advancements may not directly reach end consumers or store shelves; they can target larger enterprises, serving as suppliers of new raw materials. These materials undergo stabilization, formulation, and commercialization in various sectors, including export markets. Simplifying the understanding of these clients' needs and developing waste-derived products with desired quality and performance, alongside identifying niche markets, can help to mitigate risks and uncertainties [125].

In Rio de Janeiro, the proximity to local, regional, and overseas post-processing industries, such as food and cosmetics, which are also exporters, brings a rush for new products and, consequently, a drive for innovation that motivates new developments and the search for new biotechnological materials. These chemical entities are, on the one hand, already known and have well-defined functional properties, but on the other hand, they find competitive prices and green sales approaches by exploring residual biomass.

*4.3. Technology*

The valorization of agroindustrial waste through the extraction of bioactive compounds faces significant technological challenges, as highlighted in recent studies of phenolics [128–131]. One of the main hurdles is the development of efficient and sustainable extraction methods that can operate within the framework of a circular bioeconomy, optimizing the recovery of valuable compounds while minimizing the environmental impact. Some alternatives, such as the utilization of natural deep eutectic solvents (NADESs) based on choline chloride, represent innovative technologies offering a promising alternative to conventional methods. They overcome challenges related to extraction efficiency and the environmental sustainability of the traditional organic solvents, such as petrol ether and chloroform. These solvents, specially formulated from choline chloride and lactic acid, exhibit the exceptional ability to extract specialized plant metabolites, achieving extraction percentages exceeding 70% [128,129]. However, like any innovative technology, they present additional technological challenges, such as increased viscosity and difficulties in solvent removal. Similar drawbacks were observed with new solvents, like Cyrene®, a biocompatible, non-toxic solvent obtained via semi-synthesis from biomass, and a friendly alternative to green solvents, such as *N*-methyl-2-pyrrolidone and dimethylformamide. Unfortunately, the Cyrene® boiling point of 227 degrees makes its use very difficult due to the need for a high vacuum to remove it, even higher than the other two substances, 202 °C and 153 °C, respectively [132].

The selection of the most suitable extraction and concentration methods depends heavily on the specific type of agroindustrial waste being processed and the desired substance, along with its required level of purity. It is crucial to note a significant paradigm shift between academia and industry; unlike allopathic medicines, which demand high purity, natural products derived from biomass often do not require concentrations exceeding 70% for industrial applications. In some cases, even concentrations as low as 0.1 to 2% suffice, as seen with tea extracts.

Ultrasound-assisted extraction has emerged as a valuable technique for recovering phenolic compounds from agri-food waste. This method harnesses ultrasonic energy to enhance solvent penetration and break down the waste matrix, facilitating the release of

desired compounds. It has shown particular efficacy in extracting antioxidants, proving to be both efficient and environmentally sustainable [128,129,131].

Supercritical fluid extraction, utilizing supercritical $CO_2$ as a solvent, has garnered attention as a green extraction method. It is well-suited for obtaining heat-sensitive compounds, offering a safe and eco-friendly alternative for extracting essential oils and lipids [130]. However, challenges, such as the high pressures required for extraction and the need for co-solvents, have limited its widespread application.

It is worth considering recent advancements in chromatographic systems, enabling the purification of substances both on an analytical scale (for identification) and a preparative scale (for obtaining purified substances). Liquid chromatography (HPLC) stands out in this regard, as previously highlighted, equipped with both traditional yet effective detectors, like UV-Vis and more modern mass spectrometry detectors. These technologies have been instrumental in precisely characterizing and quantifying extracted phenolic compounds [129,131]

It is important to note that, despite the significant advantages of green techniques in terms of sustainability and efficiency, economic viability still poses a challenge. The initial costs associated with the development and implementation of deep eutectic solvents, as well as the operational costs associated with advanced techniques, such as ultrasound-assisted and microwave-assisted extraction, can be considerably higher compared to conventional methods. Optimizing these techniques to reduce operational costs and increase the recovery of valuable compounds is crucial for their industrial-scale application. Many studies focus more on the efficiency, environmental benefits, and technical aspects of extraction techniques, without detailing the operational or implementation costs in monetary terms. This highlights a gap in the available literature, where discussions on the economic feasibility of green extraction technologies often do not include detailed cost data. Such information would be crucial for a comprehensive analysis of the industrial applicability of green technologies, considering both environmental benefits and financial costs [133–137].

The effective revaluation of these compounds into marketable products certainly demands technological innovations. Some may relate to production scales, while others address regulatory issues. It is noteworthy that major technological breakthroughs in the extraction processes themselves might not always be necessary. Many simple and widely available methodologies for ethanol extraction and the subsequent selective precipitation of fats enable the concentration of phenolic fractions in quantities suitable for their biological effects and commercial appeal. In cosmetics, for instance, a minimal amount of active ingredients is preferred to ensure gradual effects and prevent any issues of skin over-absorption. Surmounting these technological challenges entails not only scientific progress but also innovation in process engineering, the sustainable design of production systems, and the establishment of value chains conducive to a circular economy in agribusiness.

## 5. Conclusions and Perspectives

The use of agroindustrial waste has led different industries to take a slightly more careful look at their ways of operating. Excessive waste disposal eliminates different bioactive compounds of interest, which have applications in different areas of the industry itself, such as textiles, cosmetics, pharmaceuticals, and food. The State of Rio de Janeiro appears as a model site in the production of different items, such as pineapples, bananas, coffee, and citrus, among others, in addition to presenting quantitative values of waste generated. Many compounds present in these matrices are known for their activities and product improvements.

It is intrinsic to state that some challenges need to be overcome for this production to become more efficient. Several technologies are on the shelf and available, but they need several adjustments, upscaling, and actualization to greener perspectives. Together with a robust infrastructure needed to capture, treat, and convert this waste into value-added compounds, the involvement between partners from the main fruit and processed food industries and new biotechnology industries has to be stimulated. All of this involves an

investment in renewable technologies and the market acceptance of these new products. What is certain is that many important opportunities are being thrown away, and different businesses that could generate more richness are being wasted, further weakening the economy and the environment as a whole, when waste reuse is considered to be a bioeconomy only when focused on huge energy entrepreneurship.

**Author Contributions:** Conceptualization, F.K.F.D.S. and V.F.V.-J.; methodology, V.F.V.-J.; software, I.G.C.B.-S. and O.L.-B.; resources, R.R. and Y.C.-S.; data curation, F.K.F.D.S.; writing—original draft preparation, F.K.F.D.S. and V.F.V.-J.; writing—review and editing, F.K.F.D.S., O.L.-B. and R.R.; visualization, F.K.F.D.S. and R.R.; supervision, V.F.V.-J.; project administration, V.F.V.-J.; funding acquisition, V.F.V.-J. All authors have read and agreed to the published version of the manuscript.

**Funding:** CAPES (Coordination for the Improvement of Higher Education Personnel), CNPq (Brazilian National Council of Research, Grant number 310782/2022-8) and FAPERJ (Grant Numbers E-26/200.512/2023 and E-26/211.315/2021).

**Institutional Review Board Statement:** Not applicable.

**Informed Consent Statement:** Not applicable.

**Data Availability Statement:** Data are available on request.

**Conflicts of Interest:** The authors declare no conflicts of interest.

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
