# Peer review of "High Added-Value by-Products from Biomass: A Case Study Unveiling Opportunities for Strengthening the Agroindustry Value Chain"

_2673-8783, doi:10.3390/biomass4020011_

Round 1

Reviewer 1 Report

Comments and Suggestions for Authors

Comments

The perspective submitted by dos Santos et al, titled “High added-value by-products from biomass: A case study unveiling opportunities for 2
strengthening the agroindustry value chain” aims to highlight the production of value-added chemicals from the biomass. The manuscript presents an interesting topic, however, there are few concerns that deserve thorough attention before publication.

Comments

1.     Line 34; Remove the phrase “keyword”.

2.     Line 145, Correct the phrase, “restinga”.

3.     Lines 276-277; At many places, the manuscript lacks flow of information. The authors have touched a variety of topics without going deep into the scientific problem with a specific focus.

4.     Lie 380; remove “.”

5.     Line 457; provide the link for the International Coffee Organization.

6.     Lines 461-465; the data is vague without a proper reference.

7.     After every section of the manuscript, authors should precisely provide a critical analysis of the available data providing their synthesis and suggestions to be addressed in the future research.

8.     Line 571-573; the statement is repetitive and needs revision.

9.     Line 625; correct the spelling of “content”.

10.  Line 716; Revise the sentence, associated with their collection, transportation, and processing.

11.  Line 833; replace, an with a.

12.  Line 842; support your statement with proper and latest reference.

13.  Conclusions and perspectives: author should provide more details about the need in the improvement of pretreatment processes to extract maximum energy from the waste biomass.

Author Response

Dear Editor,

We would like to express our sincere appreciation for your swift attention to our manuscript, Biomass-2906976, titled "High added-value by-products from biomass: A case study unveiling opportunities for strengthening the agroindustry value chain." It is truly gratifying to witness a journal that prioritizes efficiency and expedience in its review process.

We extend our sincere gratitude to the reviewers for their meticulous analysis and constructive feedback. Their contributions have undeniably elevated the quality and clarity of our manuscript. We have carefully considered and addressed all of their comments, ensuring that each point is either attended to or justified. Below, we provide a comprehensive list of the changes made, with corresponding references to the main text where these modifications can be observed, highlighted for clarity.

With my very best regards,

Filipe.

Reviewer 1

  1. Line 34; Remove the phrase “keyword”.

Answer: The term was removed as solicited.

  1. Line 145, Correct the phrase, “restinga”.

Answer: The phrase was corrected as follows: It includes stretches of semi-deciduous seasonal forests and areas with an extensive strip of sandy deposits parallel to the beach line, characterized by a hot and humid climate, with rainy seasons in the summer and dry periods in the winter.

  1. Lines 276-277; At many places, the manuscript lacks flow of information. The authors have touched a variety of topics without going deep into the scientific problem with a specific focus.

Answer: In response to the reviewer's observations, our manuscript provides at this line (266-267) an overview of the various waste materials prevalent in the state of Rio de Janeiro, along with their potential applications, as depicted in Figure 1.C. Subsequently, commencing from line 287, we delve into detailed presentations of four of these residues, aligning with the primary objective of our study.

  1. Line 380; remove “.”

Answer: It was removed as solicited.

  1. Line 457; provide the link for the International Coffee Organization.

Answer: The following link was inserted in the respective reference number 77 (https://icocoffee.org/pt/resources/public-market-information/.)

  1. Lines 461-465; the data is vague without a proper reference.

Answer: The same reference, 77, was inserted.

  1. After every section of the manuscript, authors should precisely provide a critical analysis of the available data providing their synthesis and suggestions to be addressed in the future research.

Answer: We appreciate the reviewer's suggestion, very important. Indeed, throughout the sections, the main constituents present in the specific residues of each food matrix are presented, pointing out their large-scale concentration and real values of these compounds being sold on websites specialized in the sale of reagents and chemical substances (line 441). Furthermore, the current challenges that this new perspective faces and the impact of these actions on the market are discussed. Increasing investment in technology (lines 818-820) and qualified labor by producers, logistics optimization (722-733) and market acceptance for these sub-products (lines 781-790). The authors' suggestions are present in the conclusion, where some ideas are highlighted (line 897) from the perspective of current perspectives.

  1. Line 571-573; the statement is repetitive and needs revision.

Answer: The text was modified as solicited to a better format: “The main chemical substances that can be extracted from citrus waste to become high-value-added bioproducts are essential oils and phenolics. Among the phenolics, rutin, naringenin, narirutin and hesperidin stand out.”

  1. Line 625; correct the spelling of “content”.

Answer: The text was adjusted as solicited.

  1. Line 716; Revise the sentence, associated with their collection, transportation, and processing.

Answer: The text was modified as solicited to a better format: “Although some of these wastes contain valuable components, the costs associated with their collection, transportation and processing are high.”

  1. Line 833; replace, an with a.

Answer: The text was adjusted as solicited.

  1. Line 842; support your statement with proper and latest reference.

Answer: The referenced was added as solicited, number 132.

  1. Conclusions and perspectives: author should provide more details about the need in the improvement of pretreatment processes to extract maximum energy from the waste biomass.

Answer: The focus of the work was to point out how the waste generated by the agroindustry is wasted without detailed attention to the compounds present there. One of the greatest differential approaches of this study was precisely not to focus on energy from the waste (which is already done currently). Regarding pre-treatments, the text mentions (lines 823-825) the use of solvents with lower environmental impact but not better efficiency, such as NADES (natural deep eutectic solvents) and robust technologies in the identification and characterization of compounds (lines 857 - 861).

Reviewer 2 Report

Comments and Suggestions for Authors

The submitted manuscript offers an in-depth review of the status and challenges associated with deriving value from agro-industrial residues, focusing on examples from pineapple, banana, coffee, and citrus in the Brazilian state of Rio de Janeiro. The topic of this case study is engaging, and the review is well written, complemented by clear figures. However, there are some points for consideration:

1.          While the manuscript effectively discusses the biodiversity, climatic patterns, and cultivation belts of Rio de Janeiro, these details might overshadow the main focus on biomass utilization. To streamline the content, it is suggested to condense this section and provide a more direct link to biomass utilization.

2.          Although the manuscript presents examples of utilizing residues from various fruits, it could benefit from a comparative analysis of these biomasses in terms of their characteristics, utilization techniques, and potential applications. This would enhance clarity and facilitate a better understanding of the subject matter.

3.          The reviewer recommends expanding the concluding section to offer a more robust perspective on the field. This could involve addressing challenges, proposing solutions, outlining future research directions, and discussing the prospects for large-scale implementation. Such additions would provide valuable insights and contribute to the overall strength of the manuscript.

Author Response

Dear Editor,

We would like to express our sincere appreciation for your swift attention to our manuscript, Biomass-2906976, titled "High added-value by-products from biomass: A case study unveiling opportunities for strengthening the agroindustry value chain." It is truly gratifying to witness a journal that prioritizes efficiency and expedience in its review process.

We extend our sincere gratitude to the reviewers for their meticulous analysis and constructive feedback. Their contributions have undeniably elevated the quality and clarity of our manuscript. We have carefully considered and addressed all of their comments, ensuring that each point is either attended to or justified. Below, we provide a comprehensive list of the changes made, with corresponding references to the main text where these modifications can be observed, highlighted for clarity.

With my very best regards,

Filipe.

Reviewer 2

The submitted manuscript offers an in-depth review of the status and challenges associated with deriving value from agro-industrial residues, focusing on examples from pineapple, banana, coffee, and citrus in the Brazilian state of Rio de Janeiro. The topic of this case study is engaging, and the review is well written, complemented by clear figures. However, there are some points for consideration:

  1. While the manuscript effectively discusses the biodiversity, climatic patterns, and cultivation belts of Rio de Janeiro, these details might overshadow the main focus on biomass utilization. To streamline the content, it is suggested to condense this section and provide a more direct link to biomass utilization.

Answer: We thank the reviewer for the commentary. On the one hand, these details seek to differentiate the common use of biomass – often aimed at energy applications, from the reuse focused on high-value compounds. On the other hand, detailed information on the state's characteristics is essential to bring information into a general overview of several related tropical regions that present similar conditions and could benefit from the same waste reuse panorama.

  1. Although the manuscript presents examples of utilizing residues from various fruits, it could benefit from a comparative analysis of these biomasses in terms of their characteristics, utilization techniques, and potential applications. This would enhance clarity and facilitate a better understanding of the subject matter.

Answer: We thank again the reviewer for the commentary. Currently, the industry has been looking for better alternatives for the destination for all the waste generated. Part of these actions still aim to transform this agro-industrial waste into low-value materials, like fertilizer, animal supplementation and generation of energy by burning this biomass (lines 55-61). But the composition of these "by-products" presents a chemical richness that is indispensable in different sectors, with different applications in the beverage, cosmetics, textile and pharmaceutical industries. The pre-treatments and technologies used for this "new" concept are mentioned in lines 823-825 and 857-861, respectively.

  1. The reviewer recommends expanding the concluding section to offer a more robust perspective on the field. This could involve addressing challenges, proposing solutions, outlining future research directions, and discussing the prospects for large-scale implementation. Such additions would provide valuable insights and contribute to the overall strength of the manuscript.

Answer: We thank again the reviewer for the commentary. The text below was added at the conclusion, to a better presentation of the challenges and solutions, even though they are already addressed in item 4 of the text:

“It is intrinsic to state that some challenges need to be overcome for this production to become more efficient. Several technologies are on the shelf, and available, but they need several adjustments, upscaling, and actualization to greener perspectives. Together with a robust infrastructure needed to capture, treat and convert this waste into value-added compounds, the involvement between partners from the main fruit and processed food industries and new biotechnology industries has to be stimulated. All of this involves investment in renewable technologies and market acceptance of these new products. What is certain is that a lot of important opportunities are being thrown away and different businesses that could generate more richness are being wasted, further weakening the economy and the environment as a whole, when waste reuse is considered to be bioeconomy only when focused on huge energy entrepreneurship.”